# It’s a Trap!—Potential of Cathepsins in NET Formation

**DOI:** 10.3390/ijms262211213

**Published:** 2025-11-20

**Authors:** Pola Pruchniak, Adrianna Niedzielska, Rafał Nejfeld, Zbigniew Wyżewski, Karolina P. Gregorczyk-Zboroch, Lidia Szulc-Dąbrowska, Małgorzata Gieryńska

**Affiliations:** 1Department of Preclinical Sciences, Institute of Veterinary Medicine, Warsaw University of Life Sciences-SGGW, 02-787 Warszawa, Poland; pola_pruchniak@sggw.edu.pl (P.P.); adrianna_niedzielska@sggw.edu.pl (A.N.); rafal_nejfeld@sggw.edu.pl (R.N.); karolina_gregorczyk_zboroch@sggw.edu.pl (K.P.G.-Z.); 2Institute of Biological Sciences, Cardinal Stefan Wyszynski University in Warsaw, 01-815 Warszawa, Poland; z.wyzewski@uksw.edu.pl

**Keywords:** neutrophils, cathepsins, neutrophil extracellular traps, NETosis, innate immunity, inflammation, therapeutic targets

## Abstract

Neutrophils are first-line immune effectors in innate immunity, employing migration, phagocytosis, and neutrophil extracellular trap (NET) formation to combat infections and mediate inflammatory responses. NET formation, the regulated extrusion of chromatin and antimicrobial proteins, is crucial for pathogen clearance but can lead to pathological inflammation when dysregulated. Cathepsins, a diverse family of proteolytic enzymes traditionally associated with lysosomal protein degradation, have emerged as key modulators of neutrophil functions. Serine cathepsins, including cathepsin G, and cysteine cathepsins, such as cathepsin C, regulate neutrophil migration, chemokine processing, and serine protease maturation, thereby orchestrating effective phagocytosis and antimicrobial activity. These enzymes also influence NET formation, linking classical lysosomal proteolysis to specialized immune responses. This review synthesizes current evidence on cathepsin-mediated regulation of neutrophil effector functions, highlighting their dual role in host defense and disease pathology, and discusses their potential as therapeutic targets for mitigating NET-driven inflammation in conditions such as autoimmune diseases, cancer metastasis, and ischemia–reperfusion injury.

## 1. Introduction

Neutrophils comprise 50–70% of leukocytes within the human bloodstream [1]. They are among the first immune cell responders during the initial acute inflammatory phase triggered by infection, environmental exposure, and cancer development [2]. These cells play a central role in innate immunity by acting when inflammation occurs and subsides [3]. Beyond their sheer abundance and rapid mobilization, neutrophils perform essential anti-infective functions associated with the presence of cytoplasmic granules that harbor a broad spectrum of antimicrobial molecules, including proteases such as neutrophil elastase (NE) and cathepsin G, as well as myeloperoxidase (MPO), defensins, and lactoferrin. This set of enzymes enables them to kill pathogens efficiently engulfed during phagocytosis and to contribute to extracellular defense mechanisms such as degranulation and the release of neutrophil extracellular traps (NETs) [1,3,4].

During infection, neutrophils kill pathogens through three primary strategies: phagocytosis, degranulation, and the release of NETs [5]. NETs are the extrusion of decondensed chromatin from the cell nucleus and even mitochondria, forming fibrous weblike structures saturated with histones and antimicrobial agents derived from cytoplasmic granules [6,7]. They can be formed through a specialized cell death program called NETosis, which differs from apoptosis and other lytic cell death pathways, such as pyroptosis and necroptosis [8,9,10]. While NETs play a crucial role in immune defense, their excessive or dysregulated formation is associated with various inflammatory and autoimmune diseases, non-canonical thrombosis, and tumor metastasis [2,5,11,12,13].

The NET generation requires chromatin decondensation and DNA extrusion, facilitated by the production of reactive oxygen species (ROS) and peptidyl arginine deiminase (PAD) 4-mediated histone citrullination [5,14]. Neutrophil serine proteases (NSP) and NE translocate to the nucleus to hydrolyze histones, promoting chromatin decondensation and NET release [7,15]. However, emerging evidence suggests that cathepsins, a diverse family of proteases primarily involved in intracellular protein degradation, may also contribute to the regulation of NET formation [16].

Cathepsins have traditionally been recognized for their roles in lysosomal protein degradation and immune homeostasis, participating in antigen processing and modulating inflammatory mediators [17]. Nevertheless, recent studies suggest that certain cathepsins, such as cathepsin G and cathepsin C, may influence NET formation [13,15,16]. For instance, cathepsin G has been implicated in cleaving gasdermin D (GSDMD), a key mediator of pyroptosis, suggesting potential crosstalk between NET formation and other lytic pathways [18,19]. Additionally, cathepsin C has been shown to activate NE, further implicating cathepsins as upstream regulators of NET-associated proteases [4,17].

Given the pathological implications of dysregulated NET formation, understanding the role of cathepsins is essential for developing novel therapeutic strategies for inflammatory and autoimmune diseases. This review explores the connection between cathepsins and NETs, examines their contributions to neutrophil-mediated immunity, and discusses their potential as therapeutic targets in NET-driven pathologies.

## 2. NET Formation Is Not Equal to NETosis

There are two mechanistically distinct forms of NET formation: suicidal and vital, which were presented by Guillotin et al., as two forms of NETosis [20]. Historically, NET formation and NETosis were considered synonymous. NETosis, in particular, is typically associated with neutrophil death, but recent evidence indicates that the release of NETs does not always lead to the end of a cell’s life [21]. For this reason, NET formation is considered the safer and more appropriate term, which will be used throughout the text.

Suicidal NET formation, which is equal to NETosis, is a lytic process that culminates in death, often occurring over several hours (Figure 1). It is typically triggered by strong stimuli such as pathogens or phorbol 12-myristate 13-acetate (PMA). It is characterized by ROS production via NADPH oxidase complex (NOX), histone citrullination via PAD4, and membrane rupture mediated by GSDMD. This type also involves receptor-interacting protein kinases (RIPKs) and mixed lineage kinase domain-like protein (MLKL), which further facilitate cell lysis [20,22,23,24,25].

In contrast, vital NET formation is a rapid, non-lytic process in which neutrophils retain viability and functionality after NET release. It is commonly induced by microorganisms, activated platelets (involvement of Toll-like receptor [TLR]4), or complement components, and often proceeds independently of NOX2 activity. Without nuclear or plasma membrane rupture, nuclear or mitochondrial DNA is extruded via vesicular pathways. The cell remains capable of chemotaxis and phagocytosis afterward [5,20,23,26].

The precise mechanism underlying the initiation of NET formation remains unclear, underscoring the need for further research to elucidate this process in detail. Current evidence indicates, however, that NETs release can be induced by a wide range of stimuli, including local, high concentrations of bacterial compounds (e.g., lipopolysaccharide), fungi, immune complexes, and cytokines (e.g., chemokine (C-X-C motif) ligand [CXCL]8) [26]. These signals engage surface receptors, including TLRs, Fcγ receptors, complement receptors, and G-protein-coupled receptors (GPCRs), initiating intracellular cascades that involve calcium mobilization, ROS production, and kinase activation [5,24].

A significant feature of NET formation is chromatin decondensation, which allows the nuclear content to be expelled into the extracellular space (Figure 1). This process begins with weakening the electrostatic bond between histones and DNA, primarily mediated by the citrullination of histones, a post-translational modification catalyzed by PAD4 [26]. PAD4 converts arginine residues on histones to citrulline, reducing their positive charge and weakening their interaction with the negatively charged DNA backbone [27]. As a result, the chromatin unravels, priming it for extrusion. The histone citrullination is calcium-dependent, linking it directly to the intracellular calcium spikes that follow receptor stimulation [5]. PAD4-mediated chromatin relaxation is essential for NET formation, and its inhibition or deletion (e.g., in PAD4 knockout mice) abolishes this process [28]. Further decondensation and nuclear disruption are facilitated by the translocation of azurophilic granule proteins into the nucleus. Notably, NE degrades histones and nuclear lamins, while MPO contributes to ROS production and histone oxidation [27]. Both enzymes are part of a protein complex called azurosome, which is located in azurophilic granules, and they are released in response to NOX-mediated ROS generation. The ROS-induced disintegration of the azurosome complex allows these proteases to migrate into the nucleus, where they work with PAD4 to fully decondense the chromatin [15,29]. Once decondensed, the chromatin is either expelled directly through pore formation in the plasma membrane, facilitated by GSDMD, or packaged into vesicles that transport nuclear material to the extracellular space. NE cleaves GSDMD, enabling pore formation necessary for NET formation [24] (Figure 1).

Functionally, NETs immobilize and destroy pathogens through their structural trapping mechanism and concentrated load of antimicrobial components like histones, cathelicidins, NE, lactoferrin, lysosomes, gelatinase, defensins, and MPO. Doing so prevents the dissemination of pathogens during infection [4,22,26,30]. However, NETs are not limited to beneficial functions. Excessive or aberrant NET formation has been implicated in a broad spectrum of pathological outcomes, including the exacerbation of autoimmune diseases, chronic inflammation, tissue injury, and even tumor progression [22,24,26,31,32]. These detrimental effects highlight the dual nature of NETs, which, while essential for antimicrobial defense, can also act as potent mediators of disease under dysregulated conditions.

NETs are potent antimicrobial structures capable of combating many pathogens, particularly those that resist phagocytic killing. By ensnaring microorganisms in a mesh of chromatin and antimicrobial proteins, NETs help restrict their spread throughout tissues. Nevertheless, impaired regulation or removal of NETs can contribute to tissue damage and inflammation, resulting in adverse consequences [33,34].

## 3. Cathepsins—An Overview

Cathepsins comprise a diverse family of proteolytic enzymes, classified according to the nature of their catalytic residues into three groups: cysteine, aspartic, and serine. The cysteine cathepsins—B, L, S, C, F, H, K, O, W, V, and X(Z/P)—constitute the largest subgroup. Cathepsin D and cathepsin E are aspartic proteases, while cathepsins A and G are classified as serine proteases [35,36].

Cathepsins are initially synthesized as inactive precursors (zymogens) within the endolysosomal system and are subsequently activated in acidic environments through autocatalytic mechanisms or by other proteases [37,38,39]. Although lysosomes and endosomes are their primary activity site, cathepsins have been increasingly recognized for their functional versatility in other cellular compartments, including the cytoplasm, nucleus, and extracellular space [40]. Notably, cathepsins retain proteolytic activity across a broad pH range, enabling them to function beyond the confines of the endosomal–lysosomal system. Some cathepsins also exhibit specific proteolytic functions in non-acidic cellular environments [41,42,43,44,45]. The major functions of cathepsins are summarized in Table 1; however, this review highlights the role and functions of selected cathepsins (Table 1).

Cathepsins are produced by both immune- and non-immune cell populations, reflecting their diverse biological roles. In immune cells, such as neutrophils, macrophages, dendritic cells (DCs), and B lymphocytes, cathepsins play a crucial role in regulating key defense mechanisms against pathogens and contribute to the establishment of protective immunity. For example, neutrophils express cathepsin C (also known as dipeptidyl peptidase I, DPPI), which is indispensable for activating the antimicrobial NE and proteinase 3 (PR3), while cathepsin G is stored in azurophilic granules to contribute to antimicrobial activity [74,75]. Similarly, antigen-presenting cells (APCs) (macrophages, DCs, and B cells) utilize cathepsins such as cathepsin B, cathepsin L, and cathepsin S to process and present antigens via the MHC class II pathway, directly linking these enzymes to adaptive immunity [35,44].

In contrast, non-immune cells, including fibroblasts, epithelial cells, endothelial cells, and tumor cells, also produce cathepsins. Still, their primary roles are related to tissue remodeling, regulation of apoptosis, and disease progression. For instance, fibroblasts secrete cathepsin K, which is particularly potent in degrading collagen and contributes to both normal bone remodeling and pathological processes such as osteoporosis and arthritis [76]. Epithelial and tumor cells upregulate cathepsin B and cathepsin L, which facilitate extracellular matrix degradation, invasion, and metastasis [77,78,79].

## 4. The Involvement of Cathepsins in Neutrophil Functions

Several cathepsin family members play particularly critical roles in neutrophils, the predominant innate immune cells involved in rapid responses to infection. Neutrophils express abundant levels of serine cathepsins, such as cathepsin G, as well as cysteine cathepsins like cathepsin C, which are indispensable for the activation of downstream NSP, including NE and PR3. These proteases are stored in azurophilic granules and are released upon neutrophil activation to mediate the destruction of pathogens and modulate inflammatory signaling [74,75].

Cathepsin C acts as a master regulator of NSP maturation during granulopoiesis in the bone marrow, processing zymogens into their enzymatically active forms and ensuring neutrophils are fully equipped with their antimicrobial arsenal before entering circulation [16,51]. Functionally, active cathepsin G contributes not only to direct microbial killing but also regulates neutrophil migration through extracellular matrix remodeling and activating inflammatory pathways by processing cytokines and chemokines [44,80] (Figure 2).

Moreover, the subcellular localization of neutrophil cathepsins reflects their specialized immunological roles. The cysteine cathepsins, such as cathepsin C, are active in both endolysosomal and cytoplasmic compartments, facilitating both intracellular and extracellular proteolysis during immune responses [45,81]. These specialized localizations and functions in neutrophils underscore the evolutionary refinement of cathepsins beyond generic lysosomal roles. They dynamically regulate neutrophil-mediated host defense, inflammation, and tissue remodeling, thereby maintaining immune homeostasis and contributing to disease pathogenesis in contexts where neutrophil activity is dysregulated [82,83,84].

Cathepsins, together with other mediators, play crucial roles in primary neutrophil effector functions, orchestrating key immune processes such as migration, phagocytosis, and apoptosis. These foundational roles set the stage for understanding their emerging contributions to the formation of NET.

### 4.1. Migration

Cathepsin G plays a pivotal role in promoting neutrophil migration by amplifying the inflammatory response and recruiting neutrophils, monocytes, and APCs to sites of injury or infection. It enhances chemotactic signaling by processing chemokines, such as CXCL5 and the chemokine (C-C motif) ligand (CCL) 15, into more potent forms [85,86,87] and by converting prochemerin into chemerin. Chemerin binds to the ChemR23 receptor on APCs, initiating downstream signaling events that include calcium mobilization, inhibition of cyclic adenosine monophosphate (cAMP) accumulation, and activation of mitogen-activated protein (MAP) kinases [87,88]. However, this proinflammatory signaling can be counterbalanced by cysteine cathepsins, which can degrade specific chemokines, including those attracting T cells or have angiostatic functions [89].

Beyond chemokine processing, cathepsin G critically modulates adhesion-dependent neutrophil effector functions by promoting the clustering of CD11b/CD18 integrin. This clustering is essential for full activation of neutrophil responses at sites of immune complex deposition. Cathepsin G-deficient neutrophils exhibit impaired release of CXCL2, reduced ROS production, and defective cytoskeletal rearrangement despite normal adhesion. These defects are not due to impaired synthesis of cathepsin G but rather to a failure in secretion, and they are restored by the addition of enzymatically active cathepsin G [90]. Mechanistically, cathepsin G supports intracellular signaling through pathways involving Syk, Vav1, and Rac1, which are vital for actin remodeling and effective immune function [91].

In addition to its role in immune cell recruitment, cathepsin G also activates platelets during inflammation [92]. In the presence of neutrophils, it triggers platelet secretion and aggregation through protease-activated receptor (PAR) 4 signaling [93]. Supporting its importance in inflammatory conditions, mouse models deficient in cathepsin G exhibit reduced neutrophil infiltration, particularly in acute pancreatitis, indicating its critical role in mediating neutrophil-driven inflammation [94]. Cathepsin B contributes to neutrophil extravasation by cleaving integrins, such as CD11b/CD18, on the endothelial surface. This action facilitates neutrophil detachment from platelets and promotes transmigration across the vascular barrier [48].

In contrast, cathepsin C exerts an indirect influence on neutrophil migration by regulating the maturation of serine proteases, which are essential for neutrophil activation [95]. Specifically, it activates membrane-bound PR3, which enables the processing of interleukin (IL) 1βB and the activation of nuclear factor kappa-light-chain-enhancer of activated B cells (NF-κB) signaling pathways. This cascade stimulates the release of IL-6 and CCL3, thereby amplifying neutrophil recruitment to metastatic niches, particularly during cancer progression [16,52,96,97].

### 4.2. Phagocytosis

Beyond their role in guiding neutrophils toward inflammatory foci, cathepsins also contribute to the direct elimination of pathogens through phagocytosis. Once neutrophils arrive at the site of infection, their antimicrobial arsenal, including cathepsins, is deployed within phagolysosomes to neutralize invaders.

Phagocytosis is a cellular process that enables the uptake and elimination of microorganisms, foreign substances, and apoptotic cells. It is primarily performed by neutrophils and monocytes/macrophages, and plays a crucial role in maintaining tissue homeostasis. This process proceeds in sequential steps, beginning with particle recognition and activation of internalization mechanisms, followed by the formation of a phagosome that matures into a phagolysosome, where the ingested material is degraded using oxygen-dependent and/or oxygen-independent mechanisms [98].

Neutrophils act as frontline defenders of innate immunity, performing rapid phagocytosis during the early stages of infection [99,100]. During this process, one of the significant steps is the formation of the phagolysosome. Inside phagolysosomes, the serine proteases NE, PR3, and cathepsin G function alongside microbicidal peptides (e.g., defensins and cathelicidin), MPO, and the membrane-associated NOX to destroy internalized pathogens via the generation of ROS (Figure 3A) [99]. NE, PR3, and cathepsin G may be released into the extracellular space in their active forms during both phagocytosis and neutrophil turnover. To prevent collateral tissue damage, their proteolytic activity is tightly controlled by protein inhibitors in the extracellular and pericellular environment, particularly to avoid degradation of connective tissue components such as elastin, collagen, and proteoglycans [99,100].

Neutrophils derived from cathepsin G-deficient (*CtsG*^−/−^) mice retain normal morphology, granule content, and exhibit preserved functions such as phagocytosis, superoxide production, and chemotactic responses to complement component 5a (C5a), N-formyl-methionyl-leucyl-phenylalanine (fMLP), and CXCL8. These observations suggest that cathepsin G is either non-essential for these baseline neutrophil functions or that its roles are functionally compensated in vivo by other granule proteases with overlapping substrate specificities, such as NE or PR3 [101].

### 4.3. Apoptosis

As neutrophils complete their antimicrobial mission, the decision between survival and cell death becomes critical for inflammation resolution. Cathepsins, particularly those released during phagocytosis, also play a central role in initiating neutrophil apoptosis through tightly regulated proteolytic and oxidative mechanisms (Figure 3).

The interplay between ROS and cathepsins is one of the key determinants of neutrophil fate after phagocytosis, balancing survival and apoptosis to ensure proper resolution of inflammation [102].

During phagocytosis, neutrophils generate ROS, which induces lysosomal membrane permeabilization and leads to the release of cathepsins into the cytosol. Once in the cytosol, cathepsins cleave pro-apoptotic mediators such as BH3-interacting domain death agonist (Bid), which subsequently disrupts outer mitochondrial membranes and amplifies apoptotic signaling. This establishes a feedback loop in which ROS promotes cathepsin activity, and cathepsin-mediated mitochondrial disruption enhances further ROS production [103]. The ROS–cathepsin axis operates within a broader network of regulated cell death pathways, including necroptosis and pyroptosis, underscoring its versatile role in cell death and immune regulation [102,104,105].

The neutrophil lifespan is controlled, among others, by serine cathepsins, particularly cathepsin G, whose potentially cytotoxic activity is restrained by serpin inhibitors such as Serpinb1 and Serpinb6a to prevent premature cell death [106,107].

Cathepsin D, released from azurophilic granules, initiates apoptosis by directly activating caspase-8 [56]. In ROS-deficient neutrophils, release of cathepsin D is delayed, resulting in reduced caspase-8 activation and postponed apoptosis. Importantly, cathepsin D-induced caspase-8 activation occurs independently of death receptor pathways such as those triggered by Fas or tumor necrosis factor (TNF) but remains ROS-dependent. This activation generates a 15 kDa enzymatically active caspase-8 fragment, confirmed through substrate labeling and enzymatic assays. Once activated, caspase-8 triggers the activation of downstream caspase-3, advancing the execution phase of apoptosis. Notably, cathepsin D cannot directly activate caspase-3, confirming that caspase-8 is a necessary intermediary [41,55,56] (Figure 3B).

In addition to caspase activation, cathepsin D-mediated apoptosis involves mitochondrial events, such as the release of cytochrome c and the second mitochondria-derived activator of caspases (Smac), further amplifying the apoptotic cascade. These mitochondrial changes are both caspase- and cathepsin D-dependent. During spontaneous neutrophil apoptosis, cathepsin D is rapidly released into the cytosol via ROS-dependent granule permeabilization, a process that occurs independently of caspases and precedes mitochondrial outer membrane permeabilization [55,56,104].

While apoptosis facilitates the immunologically silent clearance of cells, neutrophils can also undergo NETosis, an alternative form of regulated cell death that actively combats pathogens. Intriguingly, many molecular mechanisms and proteases involved in apoptosis, such as ROS and cathepsins, are repurposed in the orchestration of NET formation.

## 5. The Involvement of Cathepsins in NET Formation

Cathepsins function as pivotal modulators of neutrophil responses, with roles that span migration, microbial killing, and modulation of inflammation. Their activity is closely tied to granule dynamics, oxidative signaling, and protease cascades—mechanisms that are also co-opted during NET formation [89,99,100,102]. Significantly, proteases such as cathepsin G, cathepsin B, and cysteine cathepsins contribute to key steps of NET generation, including chromatin decondensation and membrane rupture [5,14,16,49]. As such, the pleiotropic functions of cathepsins in earlier neutrophil processes establish a mechanistic foundation for understanding their emerging roles in the execution and regulation of NET formation.

### 5.1. The Dependence of NET Formation on Cathepsin G and Platelets

Beyond their well-known role in homeostasis, platelets also play an active regulatory role in inflammatory processes. They influence immune responses triggered by both tissue injury and the presence of pathogens [108]. Platelets participate in the formation of aggregates with leukocytes, enhancing the migration and activation of immune cells at the inflammation sites. In addition, they release proinflammatory mediators stored in their granules, such as cytokines, chemokines, and enzymes, including cathepsins, which modulate the course of the inflammatory response [109,110].

Among the endogenous signals promoting NET formation, cathepsin G modulates platelet function that may indirectly enhance NET formation. In vitro studies show that cathepsin G mediates interactions between neutrophils and platelets, promoting cellular aggregate formation and initiating thrombus development [111]. Furthermore, neutrophil-enhanced platelet aggregation was diminished upon pharmacological inhibition of cathepsin G [92]. When examining in vitro platelet activation by neutrophil-derived proteases, cathepsin G was found to be responsible for the activation of purified human platelets. Depending on the used concentration, activation occurred via PAR-1 or PAR-4 receptors (Figure 4). However, when cathepsin G was added to whole blood at low concentrations, it strongly activated platelets through an ADP-dependent, PAR-independent mechanism [62,93]. Studies also show that NETs can directly activate platelets by inducing their aggregation, degranulation (such as release of ADP and soluble P-selectin), high mobility group box 1 (HMGB1) protein release, phosphatidylserine (PS) exposure, and glycoprotein IIb/IIIa (GP IIb/IIIa) activation, independently of thrombin or other coagulation factors [112,113,114]. Neutrophils, NETs, and IL-17 are associated with the organization of thrombi in acute myocardial infarction [113]. Inhibition of cathepsin G significantly reduced P-selectin presence, GP IIb/IIIa conformational changes, and PS exposure, indicating that cathepsin G is a key mediator of NET-induced platelet activation and may contribute to thrombosis [92,115]. Soluble P-selectin released from activated platelets enhances the presence of its ligand, P-selectin glycoprotein ligand-1 (PSGL-1) on the surface of neutrophils and increases phosphorylation of the Syk kinase, which promotes NET formation through the PSGL-1/Syk/Ca^2+^/PAD4 signaling pathway [116]. Hypothetically, Syk kinase may also activate TAK1 and phosphoinositide 3-kinase (PI3K), leading to degranulation and the release of cathepsin G from azurophilic granules [117]. HMGB1 released from activated platelets may bind to the receptor for advanced glycation end-products (RAGE) on neutrophils and activate a NOX-dependent pathway, while GP IIb/IIIa, upon binding to solute carrier family 44 member 2 (SLC44A2), may activate a Ca^2+^ flow- and NADPH oxidase-dependent pathway [118].

Beyond its indirect role in modulating NET formation through platelet activation and intercellular signaling, cathepsin G also acts directly within neutrophils to promote NET release at the molecular level. At the molecular level, cathepsin G, alongside enzymes such as azurocidin 1, NE, and MPO, is released during NOX-dependent NET formation, indicating its direct participation in the process [29]. However, cathepsin G is also likely involved in the induction of NET release itself. Supporting evidence comes from studies examining the effect of recombinant human serpin B1 (rhsB1) on NET formation, suggesting a role for human cathepsin G in NET production induced by immune complexes (ICs). In these studies, only variants of rhsB1 targeting proteases with chymotrypsin-like activity, including cathepsin G, were found to inhibit NET formation effectively. The efficacy of rhsB1 depended on its specificity and the timing of administration; only pre-treatment with the serpin inhibitor prevented NET formation. This suggests that rhsB1 modulates the response of resting neutrophils at the cell surface level [117]. Further confirmation of cathepsin G’s role in NET formation came from studies using other neutrophil protease inhibitors, such as secretory leukocyte protease inhibitor (SLPI). In this case, SLPI variants that did not inhibit NE still reduced NET formation, indicating the significance of cathepsin G or other proteases with similar specificity [117]. The importance of cathepsin G was also highlighted in studies on vaccine-induced immune thrombotic thrombocytopenia (VIITT), where its presence was significantly elevated in immunologically derived thrombi compared to non-immune-related clots. In this context, cathepsin G contributed to NET formation driven by platelet activation, indicating a key role in immune-dependent thrombotic mechanisms [60]. Additionally, studies revealed that cathepsin G does not induce NET formation via the PAR2, unlike trypsin and kallikrein 14 (KLK14). Cathepsin G cleaves PAR2 at non-canonical sites, leading to receptor inactivation rather than activation [119].

In addition to its direct role in NET formation, cathepsin G relies on upstream enzymatic activation during neutrophil maturation. This activation is orchestrated by cathepsin C, which governs the functional availability of several serine proteases essential for NET release.

### 5.2. The Dependence of Neutrophil Activity on Cathepsin C

As mentioned earlier, cathepsin C plays a crucial role in regulating neutrophil activity, particularly by influencing the process of NET formation. As an enzyme responsible for the maturation of NSPs, cathepsin C converts their inactive forms into active ones by cleaving dipeptides from the N-terminal end during neutrophil differentiation in the bone marrow [51]. Its importance is supported by clinical observations showing that cathepsin C deficiency in humans leads to significantly reduced levels and activity of key NSPs: cathepsin G, NE, and PR3 [54,120]. The link between cathepsin C and NET formation has been demonstrated in studies of Papillon-Lefèvre syndrome (PLS), which results from inactivating mutations in cathepsin C. In patients with PLS, there is a lack of typical azurophilic granule proteases, including NE, cathepsin G, and PR3, leading to impaired NET formation in response to ROS [121,122]. Although in cases of rare mutations in the cathepsin C gene, PMA-induced NET formation was limited and delayed, and it was not completely abolished [123]. Further studies have demonstrated that cathepsin C is directly involved in the induction of NETs through the activation of the NOX-dependent pathway and the production of ROS. Cathepsin C activates the p38 MAPK kinase, leading to the phosphorylation of the cytosolic p47phox subunit of NADPH oxidase. This facilitates the translocation of cytoplasmic subunits to the cell membrane and the activation of the enzymatic complex (Figure 5). Simultaneously, cathepsin C increases the expression and activity of membrane-bound PR3, leading to the enhanced release of proinflammatory factors characteristic of NET formation [53]. In the context of cancer, studies have shown that cathepsin C secreted by breast cancer cells promotes lung metastasis by regulating neutrophil recruitment and NET formation. Cathepsin C enzymatically activates membrane-bound PR3, allowing for the processing of IL-1β and the activation of the NF-κB transcription factor [52]. As a result, the synthesis of IL-6 and CCL3, cytokines responsible for neutrophil recruitment, increases.

Additionally, the cathepsin C-PR3-IL-1β axis stimulates p38 phosphorylation and ROS production in neutrophils, leading to NET formation, which degrades thrombospondin-1 and supports tumor cell growth in the lungs [52]. Thus, cathepsin C promotes NET release by regulating the IL-1β-p38-ROS axis in neutrophils, and its pharmacological inhibition shows promising therapeutic potential, particularly in conditions involving NET-related tissue damage, such as lung ischemia/reperfusion injury [52,53]. While cathepsin C acts early to regulate neutrophil granule proteases, lysosomal cathepsins such as B and D contribute to NET formation by modulating signaling downstream of metabolic and autophagic stress [49]. These cathepsins add another layer of regulatory complexity to the NET release pathway.

### 5.3. The Involvement of Cathepsins B and D in NET Formation

Autophagy is an adaptive mechanism activated in response to various stressors, such as nutrient deprivation, lack of growth signals, infection, or hypoxia [124]. The process begins with the sequestration of a portion of the cytoplasm (containing, e.g., damaged mitochondria or protein aggregates) by a membrane, leading to autophagosome formation. Autophagosomes subsequently fuse with lysosomes, which contain acidic hydrolases. Within the formed autolysosomes, the sequestered cytoplasmic components are degraded into simple metabolites, which can be reutilized by the cell [124,125].

Cathepsins, specifically cathepsin B and cathepsin D, are lysosomal cysteine proteases that play crucial roles in sphingosine-1-phosphate (S1P)-triggered NET formation. During autophagy inhibition, these proteases are diminished, which in turn influences NET formation. When autophagy is inhibited using bafilomycin A1 (Baf A1), which suppresses autophagosome-lysosome fusion, both wild-type (WT) and carcinoembryonic antigen-related cell adhesion molecule 1 knock-out (CC1-KO) neutrophils exhibit enhanced NET formation, suggesting autophagy prevents S1P-induced NET release. Baf A1 treatment depletes cathepsin B and cathepsin D, but sphingosine-1-phosphate receptor 2 (S1PR2) ligation can restore cathepsin D more efficiently in WT neutrophils compared to CC1-KO neutrophils. Furthermore, S1PR2 ligation in WT neutrophils suppresses H3Cit and p62 in the presence of Baf A1, restoring cathepsin D levels. Although S1PR2 ligation fails to restore cathepsin B in S1P-stimulated neutrophils, inhibition of cathepsin B accelerates S1P-induced NET formation in both WT and CC1-KO neutrophils. These findings suggest that cathepsin D and B play crucial roles in regulating S1P-triggered NET release [49].

## 6. Pathological Consequences of Cathepsin-Mediated Formation of NETs

While NETs play a crucial role in host defense by capturing and neutralizing pathogens, accumulating evidence indicates that their dysregulated formation can also exacerbate the progression of various pathological conditions. In autoimmune diseases such as systemic lupus erythematosus (SLE) and rheumatoid arthritis (RA), NETs expose nuclear antigens that trigger the production of autoantibodies. In thrombotic conditions, they interact with platelets and fibrin to promote immunothrombosis, contributing to the development of venous thrombosis, stroke, and myocardial infarction [22,26,32]. NETs also exacerbate tissue injury in acute respiratory distress syndrome (ARDS) and coronavirus disease 2019 (COVID-19), where they have been associated with cytokine storms and microvascular occlusion [24,31]. Importantly, cathepsins have been shown to differentially contribute to the dysregulation of NET formation, thereby promoting or amplifying pathological consequences.

### 6.1. Enhancement of Inflammation

Cathepsins play an essential role in amplifying the inflammatory response at multiple levels (Figure 6). As mentioned earlier, cathepsin C activates key NSPs, including NE, PR3, and cathepsin G. These enzymes serve as potent mediators of cytokine release and leukocyte recruitment, thereby driving the progression of the inflammatory cascade [16,51]. NET formation is closely associated with the generation of ROS, activation of the NOX, and nuclear translocation of PAD4. PAD4-mediated citrullination of histones promotes chromatin decondensation, a prerequisite for NET formation. Cathepsin C indirectly facilitates this sequence by activating neutrophil granule enzymes [16,53]. NETs consist of depolymerized DNA chromatin saturated with histones and various neutrophil granule proteins, including MPO, PR3, lactoferrin, lysozyme, gelatinase, and cathelicidin LL-37. Many of these molecules, including citrullinated and carbamoylated peptides, are recognized as autoantigens by the immune system in autoimmune diseases [126,127].

### 6.2. Thrombosis

Thrombosis is increasingly recognized as a process that extends beyond the classical coagulation cascade, integrating immune and inflammatory responses [90]. In this context, non-canonical thrombosis refers to thrombotic processes arising from immune-mediated mechanisms, particularly NET formation and the activity of neutrophil-derived proteases such as cathepsin G, rather than the traditional coagulation cascade [62,128,129].

Cathepsin G has emerged as a key modulator of NET-dependent thromboinflammation. It contributes to NET formation by promoting chromatin decondensation and enhancing neutrophil–platelet interaction [60]. In experimental thrombosis models, pharmacological inhibition of cathepsin G reduced NET release and thrombin generation, highlighting its role as an amplifier of immunothrombosis [60]. Furthermore, cathepsin G enhances platelet activation by releasing adenosine diphosphate from erythrocyte membranes, thereby promoting coagulation independently of the canonical cascade [62].

Importantly, NETs act as a prothrombotic scaffold, capturing platelets and fibrin and supporting in situ coagulation [130]. These findings underscore the concept of immunothrombosis, thrombosis driven by innate immune mechanisms rather than the classical coagulation system, as a critical component of cardiovascular and thrombotic pathology [130].

### 6.3. Tissue Injury

Uncontrolled or excessive NET release contributes to significant tissue damage (Figure 6). Various components of NETs, including cathepsins (such as cathepsin G), histones, and NETosis-associated ROS, have been shown to exert direct cytotoxic effects on host tissues [75,131,132]. Antimicrobial peptides and proteases embedded in NETs can damage cell membranes, leading to cell death in organs such as the lungs, kidneys, and liver [131]. Serine proteases released during NET formation, such as cathepsin G, degrade structural proteins, including elastin and collagen, leading to extracellular matrix breakdown and organ dysfunction [75].

Moreover, excessive or dysregulated NET formation can contribute to tissue damage and inflammation-related diseases. In myocardial ischemia–reperfusion injury, NET formation has been linked to microvascular thrombus formation, endothelial dysfunction, and the activation of inflammatory cascades, particularly in the early stages of reperfusion [133]. During lung ischemia–reperfusion injury, cathepsin C, derived from alveolar macrophages, induces NET release through the activation of p38 MAPK and NOX pathways, thereby intensifying inflammation and contributing to primary graft dysfunction [53]. Notably, cathepsin G has been implicated in the sustained tissue damage that follows ischemic events [134].

### 6.4. Tumor Progression and Metastasis

Cathepsin-mediated NET formation is increasingly recognized as a critical factor in cancer progression (Figure 6). Within the tumor microenvironment, cathepsin C activates membrane-bound PR3 on neutrophils, facilitating the processing of IL-1β and NF-κB activation. This leads to an increased secretion of IL-6 and CCL3, promoting the chemoattraction of neutrophils to metastatic niches. The PR3-IL-1β-NF-κB axis, driven by cathepsin C, also contributes to p38 activation and ROS generation, leading to NET formation to support the metastatic growth of tumor cells. NETs, in turn, enhance tumor growth by degrading antiangiogenic molecules such as thrombospondin-1, thereby fostering angiogenesis and vascular remodeling [52].

In in vitro studies on hepatocellular carcinoma (HCC), cathepsin G released during NET formation was shown to enhance cancer cell invasiveness by downregulating adhesion molecules and increasing the synthesis of proinflammatory mediators. It was also observed that the co-localization of cathepsin G with citrullinated histone H3 (CitH3), a marker of NETs, is essential for promoting HCC cell invasion. In vivo studies further confirmed the presence of this co-localization in peritumoral tissues, and its high frequency correlated with poor prognosis, indicating that cathepsin G represents a key signaling molecule in NET-driven metastasis [61]. Interestingly, studies on colorectal cancer (CRC) have shown that the expression of cathepsin G was inhibited as the tumor developed. High levels of cathepsin G correlated with better prognosis. Additionally, in vitro and in vivo studies have shown that overexpression of cathepsin G markedly suppressed viability and promoted the apoptosis of CRC. However, those studies were not linked to the NET formation mechanism [135].

In breast cancer, tumor-secreted cathepsin C enhances neutrophil recruitment and NET formation via inflammatory pathways such as the PR3–IL–1β–NF–κB axis, thereby promoting lung metastasis [52]. NETs not only contribute to inflammatory signaling but also play a mechanical role in the progression of metastasis. They can physically trap circulating tumor cells, while cathepsin G induces the release of insulin-like growth factor 1 (IGF-1), which enhances E-cadherin-mediated intercellular adhesion. This facilitates tumor cell aggregation and intravasation, leading to metastasis [2,136,137].

Studies investigating the role of neutrophils in tumorigenesis have shown that NET formation contributes to the induction of tumor necrosis. Intravascular aggregates of neutrophils and NETs obstructed the lumen of blood vessels, leading to hypoxia and necrosis, which in turn promoted metastasis. Inhibition of NET formation reduced the extent of tumor necrosis and the number of lung metastases. However, the researchers did not examine the impact of cathepsins on NET formation [138].

Together, current findings suggest that cathepsin-dependent NET formation serves not only as a disease biomarker but also as an active effector contributing to the propagation of inflammation, tissue injury, and tumor progression.

## 7. Future Directions in Cathepsin-Targeted Therapies for NET-Associated Diseases

Cathepsins have emerged as promising therapeutic targets due to their crucial role in regulating NET formation, inflammation, and tissue damage in various pathological conditions. Among these, cathepsin G and cathepsin C are particularly significant for their involvement in NET formation and disease progression. In HCC, NET-associated cathepsin G notably enhances tumor cell invasion and metastatic potential by downregulating E-cadherin level. In vitro and in vivo studies demonstrate that pharmacological inhibition of cathepsin G prevents NET formation and effectively suppresses the neutrophil-driven invasion of HCC cells [61].

Similarly, cathepsin C plays a critical upstream role by activating NSPs, including cathepsin G, NE, and PR3, which are essential for chromatin decondensation during NET formation. Inhibition of cathepsin C with small-molecule inhibitors (e.g., AZD7986) has significantly reduced NET formation and tissue injury in lung ischemia–reperfusion (I/R) models, suggesting its potential as a therapeutic target for primary graft dysfunction (PGD) following lung transplantation [53].

Moreover, treating neutrophils cultured in tumor cell-conditioned medium with Sivelestat (a PR3 inhibitor), Cl-amidine (a PAD4/histone citrullination inhibitor), or DNase I (which degrades NETs) effectively blocked NET formation and cathepsin C activity. This further underscores the interdependence between NET components and cathepsin activation [52,139]. Cathepsin C-induced NET formation was also inhibited by blocking IL-1β signaling, p38 MAPK, and ROS, highlighting the potential of targeting this pathway in inflammatory and malignant conditions [52]. Notably, pharmaceutical inhibition of cathepsin C has also been shown to suppress NE and PR3 activity in models of vasculitis, resulting in reduced NET burden and decreased disease severity in MPO-ANCA-associated vasculitis (MPO-AAV), a neutrophil-driven autoimmune disorder [16].

Despite these promising findings, several limitations and challenges must be addressed before cathepsin-targeted therapies can be implemented in clinical practice. A key issue is the functional redundancy among NSPs, such as NE, PR3, and cathepsin G, which may compensate for one another when only one is inhibited, potentially reducing therapeutic efficacy [100]. Furthermore, because cathepsin C is a master activator of these proteases, its inhibition could impair essential host defense mechanisms, especially in transplant or immunocompromised settings where neutrophil function is critical [53]. In HCC models, the in vivo effectiveness of cathepsin G inhibition was less pronounced than in vitro, suggesting that compensatory pathways or the tumor microenvironment may limit the clinical impact of cathepsin G-targeted approaches [61]. Additionally, while agents like AZD7986 and Sivelestat have shown benefits in reducing NET formation, they face challenges related to pharmacokinetics, off-target effects, and the timing of administration [52,53]. The research directions regarding immunotherapies are summarized in Table 2.

Future therapeutic strategies should focus on developing highly selective and bioavailable inhibitors that can modulate cathepsin activity without adversely affecting immune responses. Structural and enzymatic studies may help refine inhibitor specificity through techniques such as FRET-based substrate profiling [100]. Additionally, a combination of therapies that target both upstream activators (e.g., IL-1β, ROS) and downstream NET formation mediators could enhance therapeutic outcomes, particularly in complex conditions such as autoimmune vasculitis or metastatic cancer [16,52]. Integrating cathepsin inhibitors into personalized medicine frameworks, based on disease stage, immune status, and inflammatory context, will be essential for their successful clinical application.

## 8. Conclusions

NET formation plays a dual role in host defense and pathology. Over the past decade, cathepsins have garnered increasing attention as essential modulators of neutrophil function, extending beyond their traditional roles in lysosomal activity to regulate key processes, including chemotaxis, phagocytosis, apoptosis, and NET release.

This review highlights the distinct roles of various cathepsins, including cathepsin G, cathepsin C, cathepsin B, and cathepsin D, in shaping the molecular and functional components of NET formation. Growing evidence suggests that targeting cathepsins could be a therapeutic strategy to reduce NET-driven pathology in conditions such as cancer metastasis, autoimmune vasculitis, and ischemia–reperfusion injury. However, translating these findings into clinical practice presents challenges, including the redundancy of proteases, tissue-specific effects, and the need to preserve protective immune functions.

As our understanding of cathepsin-dependent NET formation evolves, new opportunities arise to develop selective inhibitors, optimize delivery methods, and customize interventions for specific diseases. Cathepsins, once viewed primarily as degradative enzymes, are now recognized as key players at the intersection of neutrophil biology and therapeutic innovation, providing a promising target for balancing host defense with immune resolution.

## Figures and Tables

**Figure 1 ijms-26-11213-f001:**
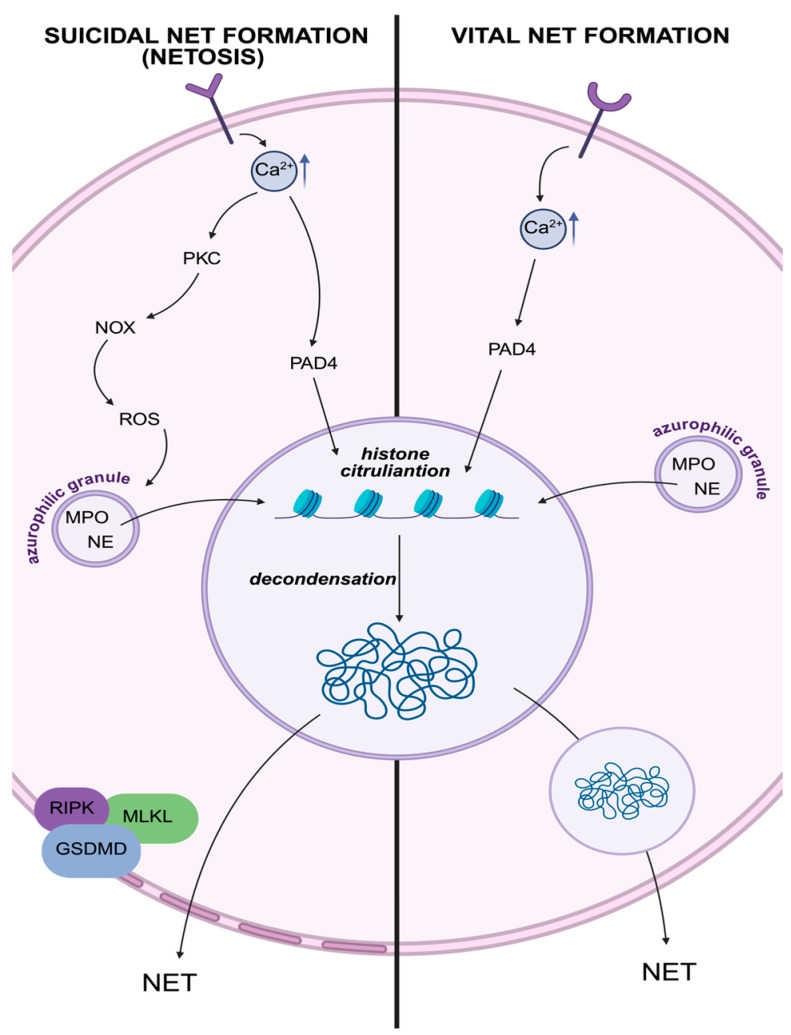
**Mechanistic comparison of suicidal and vital NET formation in neutrophils**. This figure illustrates the divergent signaling cascades and cellular outcomes underlying the two modes of NET formation. Both forms of NET release are initiated by receptor-dependent calcium influx, which activates peptidylarginine deiminase (PAD) 4, driving histone citrullination and chromatin decondensation. In NETosis (suicidal form), stimuli such as pathogens or phorbol 12-myristate 13-acetate (PMA) induce NADPH oxidase (NOX)-mediated ROS production, which facilitates the release of neutrophil elastase (NE) and myeloperoxidase (MPO) from azurophilic granules into the nucleus. These enzymes, in concert with PAD4, promote chromatin decondensation. Execution of suicidal NETosis involves membrane rupture mediated by gasdermin D (GSDMD) and kinases such as receptor-interacting protein kinase (RIPK) and mixed lineage kinase domain-like pseudokinase (MLKL), culminating in cell lysis and extrusion of NETs. In contrast, vital NET formation is a non-lytic, rapid process in which chromatin is released via vesicular transport without compromising the integrity of either the nuclear or plasma membranes. NE and MPO are similarly mobilized, and the neutrophil retains essential functions such as chemotaxis and phagocytosis post-NET release. The figure was created in BioRender. Niedzielska, A. (2025) https://BioRender.com/gjxld92.

**Figure 2 ijms-26-11213-f002:**
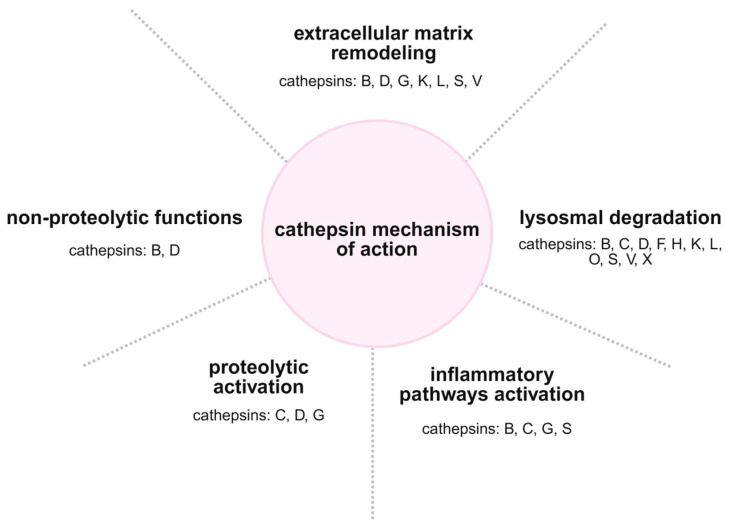
**Cathepsin mechanisms of action**. Cathepsins exert pleiotropic biological functions, including lysosomal degradation and extracellular matrix remodeling, as well as the activation of inflammatory pathways and other proteolytic enzymes. Beyond their classical proteolytic activity, they may also perform non-proteolytic functions, underscoring their complex impact on cellular homeostasis and immune responses. The figure was created in BioRender. Niedzielska, A. (2025) https://BioRender.com/mkwthff.

**Figure 3 ijms-26-11213-f003:**
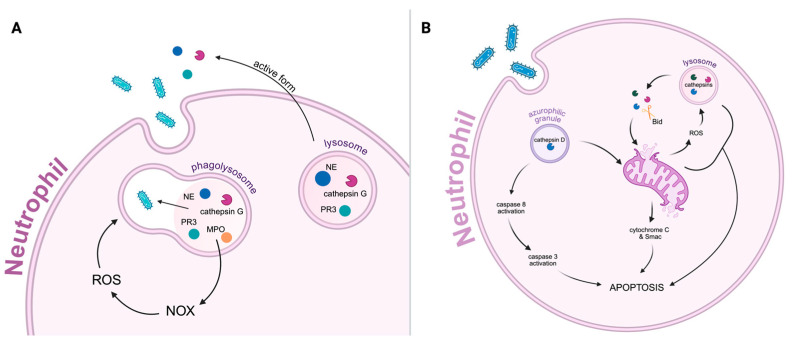
(**A**) **Schematic representation of an activated neutrophil performing phagocytosis**. The cell engulfs the pathogen, forming a phagosome that fuses with a lysosome to create a phagolysosome. Within this compartment, proteolytic enzymes such as neutrophil elastase (NE), proteinase 3 (PR3), cathepsin G, and myeloperoxidase (MPO) accumulate. These enzymes contribute to the degradation of the internalized bacterium and activate the NADPH oxidase (NOX) complex, which generates reactive oxygen species (ROS) that further enhance microbial killing. Simultaneously, some of the serine proteases (NE, PR3, cathepsin G) may be released into the extracellular space in their active forms. (**B**) **Schematic representation of neutrophil apoptosis mechanisms induced by cathepsins and ROS following phagocytosis**. Cathepsin D, released from azurophilic granules, initiates apoptosis by activating caspase-8, leading to subsequent caspase-3 activation and mitochondrial damage. Simultaneously, cathepsins released from lysosomes mediate the proteolytic cleavage of BH3-interacting domain death agonist (Bid), resulting in mitochondrial outer membrane permeabilization and the release of cytochrome c and Smac—key mediators of the intrinsic apoptotic pathway. Mitochondrial damage further enhances ROS production, amplifying pro-apoptotic signaling. The figure was created in BioRender. Niedzielska, A. (2025) https://BioRender.com/ov2ah6s.

**Figure 4 ijms-26-11213-f004:**
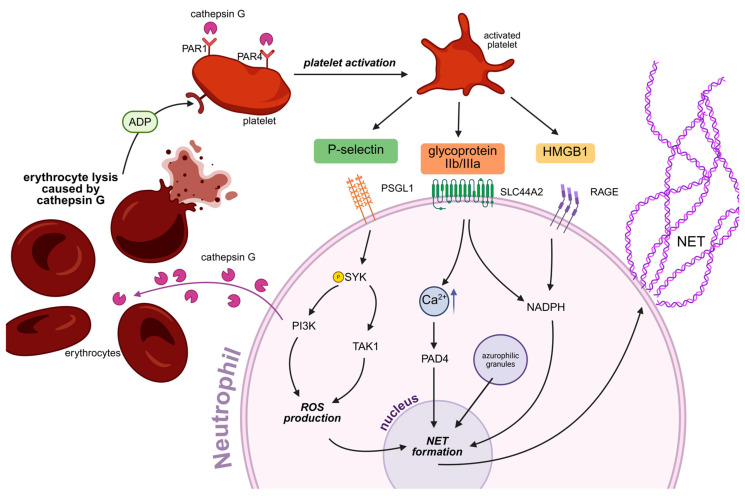
**The role of platelets and cathepsin G in neutrophil activation and NET formation induction.** Cathepsin G, released by neutrophils, can activate platelets both directly, via PAR-1 and PAR-4 receptors, and indirectly, through erythrocyte lysis and subsequent ADP release. Activated platelets secrete signaling molecules, such as P-selectin, glycoprotein IIb/IIIa (GP IIb/IIIa), and high mobility group box 1 (HMGB1), which interact with receptors on the neutrophil surface and initiate intracellular signaling cascades that promote NET formation. Soluble P-selectin binds to PSGL-1 on neutrophils, leading to Syk kinase activation, followed by PI3K-TAK1 pathway activation and enhanced ROS production. PI3K activation also promotes the release of cathepsin G from azurophilic granules into the extracellular space. GP IIb/IIIa, through its interaction with the SLC44A2 receptor, and HMGB1, via the RAGE receptor, trigger calcium- and NADPH oxidase (NOX)-dependent pathways. The rise in intracellular Ca^2+^ activates PAD4, an enzyme responsible for histone citrullination and chromatin decondensation, crucial steps in forming NETs. The figure was created in BioRender. Niedzielska, A. (2025) https://BioRender.com/lxbkobr.

**Figure 5 ijms-26-11213-f005:**
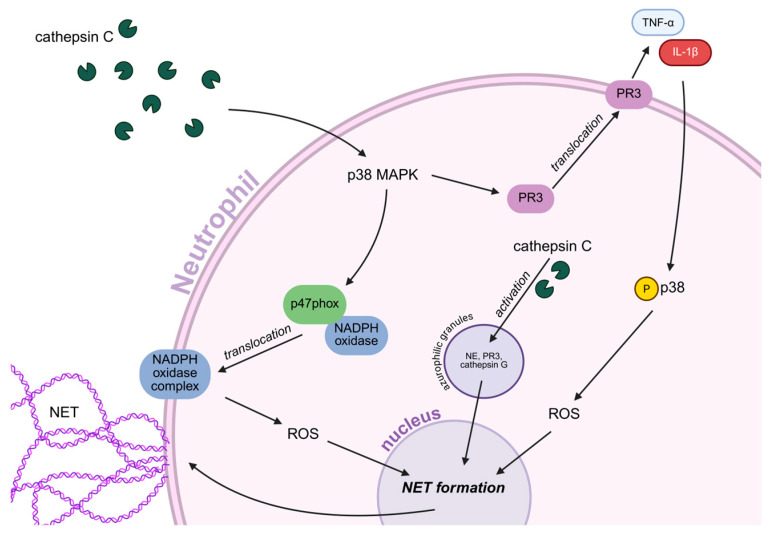
**Cathepsin C-dependent activation of neutrophils and induction of NET formation**. Cathepsin C triggers the activation of the p38 MAPK pathway, leading to phosphorylation of the p47phox subunit and translocation of the NADPH oxidase complex to the cell membrane. This produces reactive oxygen species (ROS), which serve as a key signal initiating NET formation. Simultaneously, p38 MAPK activation enhances the expression and membrane-associated activity of proteinase 3 (PR3), promoting the release of proinflammatory cytokines such as TNF-α and IL-1β. These cytokines further amplify p38 phosphorylation and ROS production in an autocrine way, enhancing NET formation. Additional internal cathepsin C activates proinflammatory serine proteases, such as neutrophil elastase (NE), cathepsin G, and proteinase 3 (PR3), which also participate in the formation of NET. The figure was created in BioRender. Niedzielska, A. (2025) https://BioRender.com/qitii4x.

**Figure 6 ijms-26-11213-f006:**
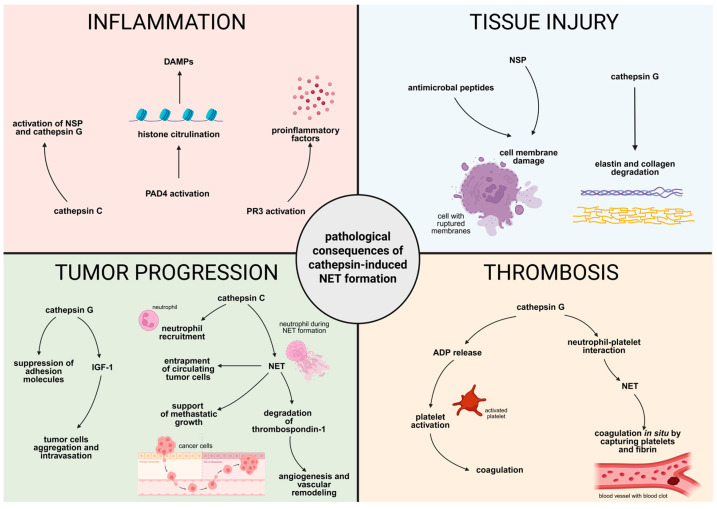
**Pathological consequences of cathepsin-induced NET formation: inflammation, tissue injury, tumor progression, and thrombosis.** Inflammation: cathepsin C activates neutrophil serine proteases (NSPs), including neutrophil elastase (NE), proteinase 3 (PR3), and cathepsin G. PAD4 activation leads to histone citrullination, enabling chromatin decondensation and the formation of neutrophil extracellular traps (NETs), which act as damage-associated molecular patterns (DAMPs); PR3 further amplifies the inflammatory response by promoting the activation of IL-1β and TNF-α, thereby intensifying the inflammatory process. Tissue Injury: excessive NET formation exerts cytotoxic effects on host tissues, cathepsin G degrades elastin and collagen, thereby contributing to the destruction of the extracellular matrix, NSPs and antimicrobial peptides embedded within NETs damage cellular membranes, leading to cell death and tissue dysfunction. Thrombosis: Cathepsin G promotes NET formation by enhancing neutrophil–platelet interactions. The resulting NETs act as a prothrombotic scaffold trapping platelets and fibrin, supporting local coagulation. Additionally, cathepsin G increases platelet activation by releasing ADP from erythrocyte membranes, further amplifying the coagulation process. Tumor progression: cathepsins play a role in the formation of a tumor-promoting microenvironment; cathepsin C enhances the IL-1β–NF-κB signaling axis, increasing IL-6 and CCL3 expression and driving neutrophil recruitment to metastatic niches. NET formation facilitates tumor progression by mechanically trapping circulating tumor cells, degrading thrombospondin-1, and promoting angiogenesis and vascular remodeling. Cathepsin G further increases tumor cell invasiveness by modulating adhesion molecules and inflammatory mediators. The figure was created in BioRender. Niedzielska, A. (2025) https://BioRender.com/y8x9syl.

**Table 1 ijms-26-11213-t001:** The major functions of cathepsins.

Cathepsin	Main Functions	References
cathepsin A	regulates blood pressure via endothelin-1 degradation; degrades bioactive peptides (bradykinin, angiotensin, oxytocin, endothelin-1); protects β-galactosidase and neuraminidase; essential for elastic fiber formation and vascular integrity	[46,47]
cathepsin B	promotes apoptosis via Bid cleavage and lysosomal membrane permeabilization; regulates autophagy-related NET formation via S1P–S1PR2 signaling; inhibition accelerates S1P-induced NETosis; facilitates neutrophil extravasation by cleaving integrins (CD11b/CD18)	[48,49,50]
cathepsin C	is a master activator of neutrophil serine proteases (NE, PR3, cathepsin G); induces NETosis via NOX-dependent ROS and p38 MAPK signaling; activates PR3–IL-1β–NF-κB axis in neutrophils, promoting NET formation; promotes lung metastasis and ischemia–reperfusion injury via IL-1β–p38–ROS pathway; mutations in the gene encoding cathepsin C cause Papillon–Lefèvre syndrome with reduced NET formation	[51,52,53,54]
cathepsin D	mediates MHC-II antigen processing and apoptosis via Bid/Bax cleavage; activates caspase-8 and caspase-3 in neutrophils during ROS-dependent apoptosis; regulates autophagy and S1P-related NET formation with cathepsin B; maintains lysosomal integrity and cell death balance	[49,55,56]
cathepsin E	regulates endosomal/lysosomal microenvironment and pH; participates in MHC-II–mediated antigen presentation; modulates macrophage and dendritic cell differentiation; is linked to inflammatory responses in skin and mucosal tissues	[57,58]
cathepsin F	is involved in invariant chain (Ii) processing and MHC-II antigen presentation; plays a role in immune regulation and inherited lysosomal diseases	[59]
cathepsin G	cleaves gasdermin D (GSDMD), linking pyroptosis and NETosis; promotes platelet–neutrophil interactions via PAR-1/PAR-4 and ADP-dependent signaling; enhances NET-driven thrombosis and tumor metastasis (HCC, breast cancer); activates chemokines (CCL15, CXCL5) to promote neutrophil migration; regulates extracellular matrix remodeling.	[18,60,61,62]
cathepsin H	participates in prohormone processing; regulates cell cycle progression; contributes to proteolytic homeostasis in endolysosomes	[63]
cathepsin K	regulates TLR signaling and β-endorphin processing in the brain; controls osteoimmune crosstalk and bone resorption; inhibition prevents inflammation-driven tissue damage	[64]
cathepsin L	regulates apoptosis and autophagy; is involved in MHC-II antigen processing and viral entry; degrades α-synuclein and tau; controls cell cycle and prohormone maturation; contributes to neuronal and aging-related degeneration.	[65,66]
cathepsin S	cleaves RIP1 kinase and regulates necroptosis; activates IL-1β and inflammasome components; participates in MHC-II antigen presentation and Li chain degradation; potential target in inflammatory and autoimmune diseases.	[67,68]
cathepsin V	is expressed in thymic epithelial cells; regulates NK and CD8^+^ T-cell differentiation; associated with epithelial tumors and immune modulation.	[69]
cathepsin W	regulates IL-2 expression in NK cells; it is a component of endoplasmic reticulum proteolytic machinery in cytotoxic lymphocytes.	[70,71]
cathepsin X/Z/P	regulates phagocytosis and intracellular protein turnover; cleaves neuron-specific enolase; mediates β_2_-integrin-dependent adhesion and T-cell migration; may participate in immune cell invasion and tissue remodeling.	[72,73]

**Table 2 ijms-26-11213-t002:** The overview of current and experimental cathepsin-targeted inhibitors, their therapeutic indications, advantages, and limitations.

Medicines	Target	Indications	Advantages	Limitations	References
Brensocatib	cathepsin C	bronchiectasis	significantly reduces the annualized pulmonary exacerbation rate	long-term safety under evaluation	[140]
β-Ketophosphonic Acid 1	cathepsin G	inflammation, tissue remodeling	reversible and competitive inhibition with high selectivity	incomplete biological characterization	[141]
JNJ-10311795	cathepsin G	inflammation	inhibitor of cathepsin G and mast cell chymase; has in vivo anti-inflammatory effects	experimental; limited translational data	[142]
BI-9740	cathepsin C	heart transplantation	reduces NE and cathepsin G activity, ameliorates histopathological injury, and limits neutrophil infiltration	preclinical stage	[143]
E-64	cathepsin K cathepsin L cathepsin S cathepsin B cathepsin H	post-ischemic intestinal inflammation	protease inhibition reduces plasma proteolytic activity and granulocyte recruitment	non-specific; limited insight into individual cathepsin roles	[144]
FUT-175	serine proteases (cathepsin-related downstream pathway)	intestinal ischemia–reperfusion injury	decreases granulocyte recruitment and inflammatory infiltration	preclinical stage; broad-spectrum protease inhibition	[144]
Ala-Hph-VS-Ph, Nva-Hph-VS-Ph	cathepsin C	neutrophil degranulation suppression and cytotoxicity	highly selective; non-toxic in vitro	limited in vivo validation; indirect neutrophil effects	[145]
K777	cathepsin B cathepsin L	inflammation	inhibits cathepsin C-dependent activation of neutrophil proteases and IL-1β	broad target profile; potential off-target risks	[146]
MOD06051	cathepsin C	MPO–AAV	reduces NSP activity (NE, PR3, cathepsin G) and NET formation, ameliorated MPO-AAV	preclinical stage	[16]
BI 1291583	cathepsin C	bronchiectasis	inhibits activation of NE, PR3, and cathepsin G; strong in vivo pharmacokinetic profile	preclinical stage	[147]
E-64c	cathepsin C	neutrophil-dominant inflammatory disorders	suppresses NE activation; optimized hydrazide-based inhibitor design	in vitro only; limited biological testing	[148]
SerpinB1, Serpin B6	cathepsin G	bone marrow neutropenia, neutrophil protection	promote neutrophil survival; prevent cathepsin G-mediated necrosis	may affect other immune cells; protease–serpin imbalance risk	[106,107]
Z-Gly Leu-Phe-CMK	cathepsin G	regulation of cathepsin G–mediated neutrophil cytotoxicity during antibody-dependent cellular cytotoxicity	potent, specific, irreversible cathepsin G inhibitor; minimal effect on NE/PR3	no pharmacological or in vivo validation	[149]
IcatCXPZ-01	cathepsin C	rheumatoid arthritis	strong anti-arthritic efficacy; reduces arthritis scores and paw swelling	preclinical stage; long-term safety not yet established	[150]

## Data Availability

No new data were created or analyzed in this study. Data sharing is not applicable to this article.

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
