# Peer review of "It’s a Trap!—Potential of Cathepsins in NET Formation"

_ijms, 2025, doi:10.3390/ijms262211213_

Round 1

Reviewer 1 Report

Comments and Suggestions for Authors

The manuscript provides a comprehensive and well-structured review of the role of cathepsins in neutrophil extracellular trap (NET) formation and their broader implications in neutrophil-mediated immunity, tissue injury, autoimmune diseases, and cancer. The review effectively synthesizes current evidence linking cathepsin activity to both the regulation of neutrophil effector functions and the pathophysiology of NET-driven diseases. The topic is timely and of high relevance, given the growing recognition of NETs as critical mediators of inflammation and immune dysregulation.

The text is clearly written and presents the concepts in a logical and accessible manner. The figures are well-designed and contribute significantly to the reader’s understanding, providing concise visual summaries of complex mechanisms. The bibliography is up-to-date and appropriately selected, covering both foundational and recent studies that support the main arguments. Overall, the manuscript successfully integrates current knowledge and identifies gaps that could guide future research, making it a valuable contribution to the field.

However, there are minor issues that should be addressed to enhance the manuscript’s clarity and coherence. In particular, the selection of disease contexts discussed across the abstract, introduction, and main text is not fully aligned. Some NET-related pathologies, such as cardiovascular diseases, are introduced early in the manuscript but are not further developed in the final section (Section 6). Additionally, several references are missing or do not cite the original source, and certain sections would benefit from the inclusion of more recent and relevant studies to provide a more comprehensive and balanced overview of the field.In summary, this is a well-written and scientifically sound review that brings new insight into an underexplored aspect of neutrophil biology. After minor revisions to address structural and consistency issues, it will make an excellent contribution to the literature on NET-driven inflammatory pathologies.

Specific minor points:

- In the abstract, the authors mention “conditions such as autoimmune diseases, cancer metastasis, and ischemia-reperfusion injury.” Similarly, at the end of the introduction, they refer to “developing novel therapeutic strategies for inflammatory and autoimmune diseases.” However, Section 6 focuses only on inflammation, tissue injury, and cancer. I suggest that the authors ensure consistency throughout the manuscript regarding the disease contexts discussed. I would also strongly recommend including a section on cardiovascular diseases (including ischemia-reperfusion injury), as these are major global health concerns and are closely linked to NET formation.

- Some references are missing. For instance, on page 2, lines 83–85 (“…but recent evidence indicates that the release of NETs does not always lead to the end of a cell’s life…”). In addition, review articles are sometimes cited instead of the original research papers. For example, on page 2, line 59, and page 4, line 153.

- Finally, I suggest that the authors consider integrating recent publications on NET formation and cancer into their review. For example, the following highly relevant article: https://www.nature.com/articles/s41586-025-09278-3

Author Response

First, we would like to thank the reviewer for the valuable comments. The manuscript was revised according to the suggestions, and the revised parts are marked in yellow.

Comment 1 However, there are minor issues that should be addressed to enhance the manuscript’s clarity and coherence. In particular, the selection of disease contexts discussed across the abstract, introduction, and main text is not fully aligned. Some NET-related pathologies, such as cardiovascular diseases, are introduced early in the manuscript but are not further developed in the final section (Section 6). Additionally, several references are missing or do not cite the original source, and certain sections would benefit from the inclusion of more recent and relevant studies to provide a more comprehensive and balanced overview of the field. In summary, this is a well-written and scientifically sound review that brings new insight into an underexplored aspect of neutrophil biology. After minor revisions to address structural and consistency issues, it will make an excellent contribution to the literature on NET-driven inflammatory pathologies.

Answer: The fragment regarding the NET pathologies was revised and improved, and additional references regarding original studies were added

Specific minor points:
Comment 2: In the abstract, the authors mention “conditions such as autoimmune diseases, cancer metastasis, and ischemia-reperfusion injury.” Similarly, at the end of the introduction, they refer to “developing novel therapeutic strategies for inflammatory and autoimmune diseases.” However, Section 6 focuses only on inflammation, tissue injury, and cancer. I suggest that the authors ensure consistency throughout the manuscript regarding the disease contexts discussed. I would also strongly recommend including a section on cardiovascular diseases (including ischemia-reperfusion injury), as these are major global health concerns and are closely linked to NET formation.

Answer: The appropriate fragments were added to the manuscript.

Comment 3: Some references are missing. For instance, on page 2, lines 83–85 (“…but recent evidence indicates that the release of NETs does not always lead to the end of a cell’s life…”). In addition, review articles are sometimes cited instead of the original research papers. For example, on page 2, line 59, and page 4, line 153.

Answer: It was corrected according to the suggestions

Comment 4: Finally, I suggest that the authors consider integrating recent publications on NET formation and cancer into their review. For example, the following highly relevant article: https://www.nature.com/articles/s41586-025-09278-3

The appropriate fragment was added with the suggested reference.

Reviewer 2 Report

Comments and Suggestions for Authors

This article is focused on the role of cathepsins in the formation of NETs. I believe that such reviews are necessary to comprehend the role of neutrophils in norm and pathology, the mechanisms of NET formation, and the creation of strategies for regulating these mechanisms.

Cathepsins are crucial in forming neutrophil extracellular traps (NETs) and mediating their pathological effects in different diseases. The interaction between cathepsins and NETs is complex, as cathepsins contribute proteolytic components that facilitate NET formation and amplify tissue damage. The role of cathepsin C in activating neutrophil serine proteases is crucial to NET formation, while cathepsin G and other cathepsins act as key effectors within formed NETs. The pathological effects of NETs are linked to diverse disease states, such as cardiovascular thrombosis, autoimmune inflammation, cancer metastasis, acute lung injury, and organ-specific damage.

The literature list contains 108 articles; in the last 5 years - 28 publications (30%); in the last 10 years - 61 publications (56%). Of course, it would be better to have more publications in the last 5 years. All the more now, the role of NET and mechanisms of its formation are intensively studied.

Although the review is comprehensive and written well, I still have some comments for the authors.

  1. Line 59. What do authors mean by the term “non-canonical thrombosis”? Indeed, the NET and the components involved in its formation in particular cathepsins affect hemostasis. However, the authors do not mention this in Chapter 6. And even in figure 6 there is no indication of the role of cathepsin-induced NET formation in thrombosis and hemostasis complications. Maybe it's worth highlighting Cardiovascular and Thrombotic Pathology.
  2. Line 143. Authors mention the azurosome complex and refer to [17], but this article doesn’t use such a term. Perhaps they meant reference [18]. The 'Azurosome complex' is rarely used term but its meaning is not described by authors.
  3. Figure 2 lacks an enumeration of specific cathepsins. Line 219. The figure 2 doesn't contain the indication on cathepsins C and G.
  4. Maybe it would be advantageous if the authors added the table containing information about the role of each cathepsin with references.
  5. In Figure 4, it could have been improved if the colors of erythrocytes and platelets were different.
  6. Despite the detailed description of the mechanisms of participation of cathepsins in the formation of NETs in Chapter 5. The involvement of cathepsins in NET formation , the authors do not explain which types of NETs are involved - vital, suicidal, etc.
  7. What is the reason for the absence of internal cathepsin C in Figure 5? This image implies that it is not involved in these processes.
  8. Line 584. The authors describe the role of cathepsin G in promoting hepatocellular cancer cell invasion. They do not mention the role of cathepsin G in colorectal cancer; however, there is data about its positive influence [doi: 10.7150/ijbs.82000]. 
  9. Perhaps it would be more informative if the authors added a table with data about current known medicines that affect cathepsins and pointed out advantages and limitations in cathepsin-targeted therapies.

Author Response

First, we would like to thank the reviewer for the valuable comments. The manuscript was revised according to the suggestions, and the revised parts are marked in yellow.

Comment 1: Line 59. What do authors mean by the term “non-canonical thrombosis”? Indeed, the NET and the components involved in its formation in particular cathepsins affect hemostasis. However, the authors do not mention this in Chapter 6. And even in figure 6 there is no indication of the role of cathepsin-induced NET formation in thrombosis and hemostasis complications. Maybe it's worth highlighting Cardiovascular and Thrombotic Pathology.

Answer: The fragment regarding thrombosis was added with the appropriate references.

Comment 2: Line 143. Authors mention the azurosome complex and refer to [17], but this article doesn’t use such a term. Perhaps they meant reference [18]. The 'Azurosome complex' is rarely used term but its meaning is not described by authors.

Answer: The references were corrected, and the fragment regarding “azurosome” was added to the manuscript

Comment 3: Figure 2 lacks an enumeration of specific cathepsins. Line 219. The figure 2 doesn't contain the indication on cathepsins C and G.

Answer: Figure 2 was corrected.

Comment 4: Maybe it would be advantageous if the authors added the table containing information about the role of each cathepsin with references.

Answer: Table 1 was added

Comment 5: In Figure 4, it could have been improved if the colors of erythrocytes and platelets were different.

Answer: The colors were slightly changed

Comment 6: Despite the detailed description of the mechanisms of participation of cathepsins in the formation of NETs in Chapter 5. The involvement of cathepsins in NET formation , the authors do not explain which types of NETs are involved - vital, suicidal, etc.

Answer: Yes, it was not indicated because it was not always clearly stated

Comment 7: What is the reason for the absence of internal cathepsin C in Figure 5? This image implies that it is not involved in these processes.

Answer: Figure 5 was corrected

Comment 8: Line 584. The authors describe the role of cathepsin G in promoting hepatocellular cancer cell invasion. They do not mention the role of cathepsin G in colorectal cancer; however, there is data about its positive influence [doi: 10.7150/ijbs.82000]. 

Answer: The appropriate fragment was added together with a reference.

Comments 9: Perhaps it would be more informative if the authors added a table with data about current known medicines that affect cathepsins and pointed out advantages and limitations in cathepsin-targeted therapies.

Answer: Table 2 was added

Round 2

Reviewer 2 Report

Comments and Suggestions for Authors

I want to express my gratitude to the authors for completing the review in accordance with the recommendations. From my perspective, this work is relevant, complete, well-designed, and structured, and will provide both interesting and useful content for the readers.